# Beyond Single Embeddings: Capturing Diverse Targets with Multi-Query Retrieval

## Abstract

Most text retrievers generate *one* query vector to retrieve relevant documents. Yet, the conditional distribution of relevant documents for the query may be multi-modal, e.g., representing different interpretations of the query. We first quantify the limitations of existing retrievers. All retrievers we evaluate struggle more as the distance between target document embeddings grows. To address this limitation, we develop a new retriever architecture, Autoregressive Multi-Embedding Retriever (AMER). Our model autoregressively generates multiple query vectors, and all the predicted query vectors are used to retrieve documents from the corpus. We show that on the synthetic vectorized data, the proposed method could capture multiple target distributions perfectly, showing 4x better performance than single embedding model. We also fine-tune our model on real-world multi-answer retrieval datasets and evaluate in-domain. AMER presents 6 and 16% relative gains over single-embedding baselines on two datasets we evaluate on. Furthermore, we consistently observe larger gains on the subset of dataset where the embeddings of the target documents are less similar to each other. We demonstrate the potential of using a multi-query vector retriever and open up a new direction for future work.

## 1 Introduction

As large language models (LLMs) have limited, out-dated parametric knowledge, augmenting knowledge at inference time by prepending retrieved documents has risen as a de facto solution (Fan et al., 2024; Gao et al., 2023). Recovering a diverse set of documents is crucial to provide comprehensive information (Xu et al., 2023), as an answer providing partial information can be technically correct yet misleading to users.

In this work, we study retrieving a diverse set of documents per query. We first analyze the behaviors of existing retrievers (Izacard et al., 2022; Yang et al., 2025b; Zhang et al., 2025; Lee et al., 2025) on datasets (Min et al., 2020; Amouyal et al., 2023) containing questions that admit multiple valid answers. Such questions arise either as a result of inherent ambiguity, or from cases where the desired answer is a list (e.g. *From which country did Seattle Storm make draft selections?* ). All retrieval models, trained to maximize the probability of obtaining a single target, show degrading performance as the distance between gold target documents belonging to the same query increases, as shown in Figure 1. We hypothesize that a single query vector, however well constructed, is insufficient to model multiple target distributions.

To address this limitation, we propose a new architecture, Autoregressive Multi-Embedding Retriever (AMER). Instead of constructing a single vector for each query, our model constructs multiple query vectors per query, autoregressively predicting query vectors one by one. Each predicted query vector is used to search the corpus to obtain a ranked list of documents, which we heuristically aggregate into a single ranked list. This approach directly addresses the limitations of single-query vector retrievers and enables retrieving diverse outputs.

We first evaluate our model on a new synthetic dataset, where each input query vector is paired with multiple target vectors. We design transformations (e.g., linear transformations and multi-layer perceptrons (MLPs)) to apply to the input query vector. The transformation yields target vectors that are far away from each other. We show that it is difficult for the single-vector retriever model

to capture all target distributions, only retrieving all the target embeddings at most 21% of the time, validating our hypothesis. AMER retrieves all targets perfectly (100% of the time) in this setting, across the diverse vectorized dataset that we created. The results indicate that the proposed multi-vector retriever is more suitable for modeling heterogeneous target distributions.

Going beyond the synthetic setting, we further evaluate our model on two real-world text retrieval datasets (Min et al., 2020; Amouyal et al., 2023). In such setting, we expect to see an improvement when the target documents form distinct clusters. We find that AMER exhibits small *average* performance gains compared to the single-query baseline in these datasets (6%, 16%), much smaller than those observed in the earlier synthetic experiments. However, our method shows more pronounced gains (8%, 180%) in a subset of evaluation data where target gold documents belonging to the same query are farther away from each other in the embedding space. This trend holds true for various base language models (LMs) we tested (e.g. Llama-3 (Grattafiori et al., 2024), Qwen-3 (Yang et al., 2025a)). Further analysis reveals that target documents belonging to the same query are substantially more similar than distractors in these datasets, surfacing the need for better benchmarks for diversity in retrieval. We will release our code publicly upon publication.

## 2 BACKGROUND: MULTI-TARGETS RETRIEVAL

**Task**   Given a corpus $D$ and a query $q$ that admits $m$ answers $\{a_1, \ldots, a_m\}$, systems should retrieve a subset $D_q$ of $D$ that covers all the $m$ targets. We assume document clusters $\{d_1\}, \ldots, \{d_m\}$, such that $\{d_i\}$ covers the answer $a_i$. Most existing dense retrievers employ a bi-encoder architecture where query and document encoders produce their respective embeddings, and document relevance is scored by the similarity between these embeddings. These models are trained using contrastive loss, pushing the query embedding closer to the ground truth document embeddings and away from the negative document embeddings. Therefore, if the answer document clusters are far from each other, it will be difficult for a single query embedding to fit to all of the clusters.

**Metric**   Typically retrieval performances are evaluated using RECALL @ $k$, which measures if the top-$k$ retrieved document set $D^k$ contains the target document. In datasets where there are multiple distinct valid answers, MRECALL @ k (Min et al., 2021) is used, which evaluates the top-$k$ retrieved document set $D^k$ against $m$ target documents.

- If $k \geq m$, then MRECALL@$k = 1$ if $D^k$ contains all $m$ answers; else MRECALL@$k = 0$.

- If $k < m$, then MRECALL@$k = 1$ if $D^k$ contains $k$ answers; else MRECALL@$k = 0$.

**Multi-answer QA Datasets**   We evaluate on two popular, easy-to-evaluate multi-answer QA datasets, AmbigQA (Min et al., 2020) and QAMPARI (Amouyal et al., 2023). We discuss other retrieval datasets with multiple targets in Appendix A.1. AmbigQA dataset contains questions sampled from Natural Questions dataset (Kwiatkowski et al., 2019) that are ambiguous and multiple disambiguated answers of

Table 1: Data Statistics. We report the number of instances in each data split, and the average number of targets per question in the test set.

|  | # Train | # Test | # Targets |
|---|---|---|---|
| AmbigQA | 5,044 | 827 | 2.58 |
| QAMPARI | 32,023 | 531 | 6.09 |

each question. Each question is thus associated with multiple target documents, which answer the question with different disambiguations.[1] QAMPARI contains questions with a list of entity answers spanning multiple paragraphs, which necessitates retrieving many documents. We present examples of each dataset in Appendix A.3. We use the Wikipedia corpus from previous work (Amouyal et al., 2023), where each passage spans 100 words on average. The corpus contains roughly 25M passages.

Table 1 presents the data statistics. The details of how we construct the training and evaluation set are in Appendix A.3. We mostly kept AmbigQA as is, and for QAMPARI, we filter the original dataset, keeping only questions with five to eight target documents for efficient development.

---

[1]AmbigQA contains some single-answer examples, which we discard. We only keep the multi-answer ones.

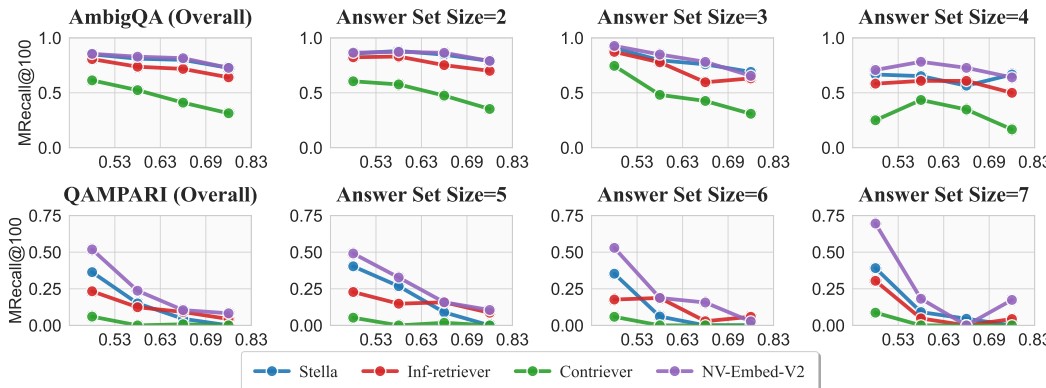

Figure 1: Model performances per the diversity of target document set. We report performance on the whole test set (the leftmost subplots), and subsets of different number of target documents (answer set size). We partition the dataset into 4 bins (<25%, 25-50%, 50-75%, and >75%) in terms of distance between target document embeddings. As the distance becomes larger, the performance worsens. The trend holds true for all models, and is more pronounced in QAMPARI dataset, where there are more answers for each query and larger distance.

## 3 FAILURE OF SINGLE-VECTOR RETRIEVERS: LIMITED PERFORMANCE ON DIVERSE TARGETS

Our key claim is that existing methods fail on examples where its multiple target documents are different from each other. We validate this by evaluating models on multi-target retrieval datasets (Min et al., 2020; Amouyal et al., 2023), checking whether the model performance degrades when the target documents are farther away from each other.

**Setting** We embed the target documents using the document encoder of the retriever, and compute the average distance (Euclidean and Cosine distance) between any two target document embeddings per query. We partition the datasets evenly into four subsets based on the average distance between all target document embedding pairs, and report the performance on each subset. We consider the target distribution more diverse when the distance between target embeddings is larger.

**Evaluated Systems** We consider four off-the-shelf retrievers, Contriever (Izacard et al., 2022), Stella (Zhang et al., 2025), Inf-Retriever (Yang et al., 2025b), and NV-Embed (Lee et al., 2025) [2]. Contriever is a commonly used compact dual encoder model trained with large-scale unsupervised contrastive learning. The other three models, all sharing similar architecture and learning objective with Contriever, are selected due to their superior performance on the MTEB benchmark (Muennighoff et al., 2023) in their respective sizes (400M, 1.5B, 7B) at the time of writing. Specifically, the latter two models are initialized from decoder-only LMs.

**Results** We present the results in Figure 1. On the left side, we report the results on the entire dataset. On the right side, we report the results while controlling for the size of answer set, potentially conflating factor (as bigger target set can make comprehensive retrieval harder).

All four models exhibit a trend that performance deteriorates as the distance between target document embeddings increases. For the AmbigQA dataset, better retrieval model (e.g., NV-Embed-V2) shows improved performance on challenging examples. Yet, on QAMPARI, Using an improved model (NV-Embed-V2) does not mitigate this issue significantly, most of the gains coming from examples where target document embeddings are closer to each other. This result reveals the limitation of existing single-query retrievers for retrieving diverse data.

---

[2]Full model names: `contriever-msmarco`, `NovaSearch/stella_en_400M_v5`, `infly/inf-retriever-v1-1.5b`, and `nvidia/NV-Embed-v2` respectively.

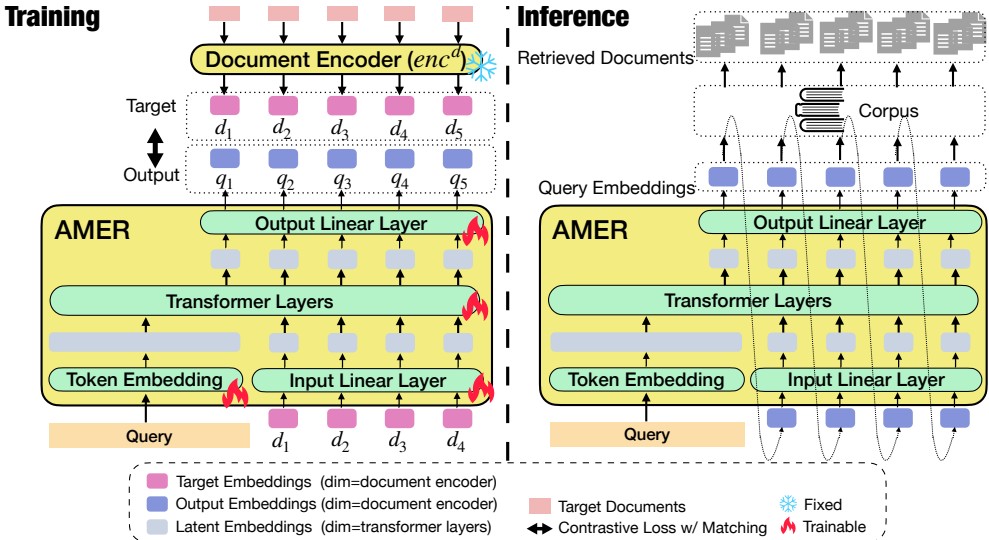

Figure 2: Visualization of AMER. We visualize both the training (left) and inference (right) procedure. The proposed model takes as input the target document embedding (order decided randomly) or predicted embedding in the previous step, and output the next embedding. Linear layers are added to ensure consistent dimensions. During inference, AMER predicts the first embedding after seeing the query text, and outputs multiple query embeddings autoregressively.

# 4 MODEL: AUTOREGRESSIVE MULTI-EMBEDDING RETRIEVER (AMER)

We investigate dense retrievers that use a common bi-encoder framework: both query and document encoders generate embeddings, whose similarity determines document relevance. Existing retriever training pipelines usually update both encoders and share their weights. However, we assume a setting where we have a frozen document encoder $\mathsf{enc}^\mathsf{d}(\cdot)$, and a query encoder $\mathsf{enc}^\mathsf{q}(\cdot)$ that is being trained. We choose this setting for faster development, as changing the document encoder requires re-generating the document indices for the entire corpus during inference.

We train a retriever that can generate multiple, distinct query embeddings. A traditional dense retriever would take the query text $q$, and output a query embedding $\boldsymbol{q} := \mathsf{enc}^\mathsf{q}(q)$. We propose to train a multi-query vector retriever such that it takes the query $q$ as input and predicts multiple query embeddings: $\mathsf{enc}^\mathsf{q}(q) := \{\boldsymbol{q}_1, \boldsymbol{q}_2, \ldots, \boldsymbol{q}_m\}$. Figure 2 visualizes our approach. AMER takes as input the query in raw text, followed by the target document embeddings. The first output embedding is produced after seeing the query text. The model then takes the target document embeddings and outputs the next query embeddings. In AMER, the query encoder is essentially an autoregressive LM, which outputs a sequence of embeddings instead of a sequence of tokens. To accommodate potential differences in embedding dimensions between the document and query encoder LMs, input and output linear layers are added to project the embeddings into a common dimensional space.

## 4.1 TRAINING

Given a set of distinct ground truth documents $\{d_1\}, \ldots, \{d_m\}$, we maximize the similarity between query embeddings $\boldsymbol{q}_i$ and the target document embeddings $\boldsymbol{d}_i = \mathsf{enc}^\mathsf{d}(d_i)$, i.e. $\max \sum_{i=1}^{k} sim(\boldsymbol{q}_i, \boldsymbol{d}_i)$. Here $sim$ denotes cosine similarity. Documents are encoded by a fixed off-the-shelf retriever $\mathsf{enc}^\mathsf{d}$. We train the model with InfoNCE loss (Oord et al., 2018), defined as:

$$l(\boldsymbol{q}, \boldsymbol{d}^+)) = -\log \frac{\exp\left(sim(\boldsymbol{q}, \boldsymbol{d}^+))/\tau\right)}{\sum_{\boldsymbol{d} \in \boldsymbol{D}_{batch}} \exp\left(sim(\boldsymbol{q}, \boldsymbol{d}/\tau\right)} \tag{1}$$

for each query embedding $\boldsymbol{q}$ and its corresponding positive document embedding $\boldsymbol{d}^+$. $\boldsymbol{D}_{batch}$ denotes all the documents embeddings in the batch, including the positive ones. For training batch size $b$ and number of answer clusters $m$, the total number of document embeddings $|\boldsymbol{D}_{batch}| = b \times m$.

**Matching Loss**   Since the set of $m$ target document embeddings is unordered, training the model on any particular ordering would push it to learn an inherently random ordering signal. Therefore, we let the model generate the $m$ output embeddings, and then calculate the loss by *optimally matching* them to the set of target document embeddings. Observe that for a single query embedding $q$, the denominator remains the same regardless of the positive document. We could thus find the exact matching of the query embeddings and the ground truth document embeddings that minimizes the loss in the batch using Hungarian matching algorithm (Kuhn, 1955). The loss of a single batch is:

$$\mathcal{L}_{batch} = \min_{p \in \mathbb{P}} \sum_{(q, d^+) \in p} l(q, d^+) \tag{2}$$

where set $\mathbb{P}$ is all the possible matchings, and each element $p \in \mathbb{P}$ contains $m$ (query embedding, target document embedding) pairs. This, intuitively, matches each generated vector with its closest gold vector. We use a standard solver to efficiently solve the optimization problem underlying Eq. 2.

**Scheduled Sampling**   During training, the retriever takes the previous ground truth document vectors as input, with the order of sequence shuffled. However, during inference, the retriever could only see its previous predictions. This poses a smaller challenge for LLMs, as the outputs are mapped to discrete tokens before being fed as inputs during inference time. In our setting, the analogous procedure would be mapping the embeddings to specific documents in the corpus and providing the corresponding document embeddings as input to the retriever. This process is expensive as the size of retrieval corpus is much larger than vocabulary size. We thus adopt scheduled sampling (Bengio et al., 2015) during training. We take the input from the target vector with probability $1-p$ and from the predicted vector in the previous step with probability $p$. We set $p$ to be $\min(0.8,$ (the number of steps trained) / (total number of steps)), with $p$ increasing from 0 linearly and clipped at 0.8.

### 4.2   Inference

During inference, AMER takes as input the query $q$, and predicts multiple query embeddings autoregressively. The retriever outputs a predetermined number of steps $(m_{pred})$,[3] and we retrieve a ranked list of documents $D_i$ from the corpus separately for each predicted query embedding $q'_i$ ($i = \{1, \ldots, m_{pred}\}$). To obtain a ranked list of $k$ documents, we take documents from the ranked list $D_i$ for each query embedding in a round-robin fashion until we reach the desired list size $k$.

## 5   Experiments: Synthetic Data

We have observed in Section 3 that single-query embedding retrievers struggle on examples where they have to retrieve diverse documents. Our claim is that using only a single query embedding is inherently limited, since they cannot model multimodal target distribution well. To test this claim, we first design a synthetic dataset, where each input is paired with a set of target outputs that are farther away from each other. We come back to report results on real-world text data in Section 6.

### 5.1   Synthetic Data Creation

We design a simple task where both the input queries and the retrieval targets are vectors. To avoid confusion with generated query embedding $q$, we refer to the queries here as "input vectors". Each input vector $x \in \mathbb{R}^d$ corresponds to $m$ ground truth vectors $\{y_i\}$, where $i = 1 \ldots m$ and $y_i \in \mathbb{R}^d$ ($d = 1024$). In this simple synthetic setting, we assume a generative story where each of the $m$ target vectors is generated by applying a predetermined fixed transformation on the input vectors. A retriever would achieve perfect performance if it can learn the transformations accurately.

**Input and target distributions** We sample the input vectors from some standard distribution, such as the Gaussian distribution $\mathcal{N}(0, I)$ or the uniform distribution; we use $k = 5$ distributions, listed in A.2. Each input vector is mapped to $m = 5$ different target vectors by the same set of five transformations. We transform each input $x$ with the equation $y_i = T_i x$, where $T_i$ is the $i$th transformation. We consider two types of transformations: (1) linear transformation matrices and (2) multi-layer

---

[3]We experiment with ways of producing varying number of embeddings, but predicting a fixed number of embeddings works best.

perceptrons (MLPs). Both are randomly initialized in a way that ensures a notion of diversity in the target vectors, i.e., the pairwise distances between gold vectors of the same query are large enough. See A.2 for the full details.

**Evaluation settings** We experiment with three settings. In the first setting (denoted *Single-in-distribution*), the training and testing input vectors are only sampled from the Gaussian distribution ($\mathcal{N}(0, I)$). In the second setting (denoted as *Multi-in-distribution*), both training and testing data share the same input distribution, where we sample $\frac{1}{5}$ of the input vectors from each of the $k = 5$ distributions. In the third setting (denoted *OOD*), we sample the training data evenly from the first four input distributions, and sample the test data only from the last input distribution. This creates out-of-distribution (OOD) queries that are unseen during training time.[4]

**Corpus** We construct a corpus by combining the target vectors from all the queries and additionally sample random negative vectors to make the corpus larger. For each output query vector, a ranked list of vectors with the highest cosine similarity to the query vector are retrieved.

## 5.2 EXPERIMENTAL SETUP

**Dataset & Metrics** We consider a total of six settings, consisting of three input distributions (*Single-in-distribution*, *Multi-in-distribution*, *OOD*), and two types of transformations (linear, MLP). We create 20k training and 1k test instances for each setting. For each setting, and we train separate model and only evaluate on its corresponding test set. The retrieval corpus is also constructed per setting. To construct the corpus, we combine all the target vectors ($20k \times 5 + 1k \times 5 = 105k$ in total), and additionally sample $95k$ random vectors where each element is sampled from a Gaussian distribution $\mathcal{N}(0, 1)$ to form a retrieval corpus of size 200k. There is no document encoder as the target data is in vector form. We report MRECALL@ 10 and MRECALL@ 100 (described in Section 2) values as our evaluation metric.

**Model Training** We use `Llama-3.2-1B-Instruct` (Dubey et al., 2024) as the base retriever model. We finetune this model using the objectives described in Section 4 for around 100k steps, using a batch size of 512. We pair one random negative vector with each of the positive vector. We consider all the other (positive and negative) vectors in the same batch as in-batch negatives, including the other target vectors in the same sequence (see Section 4.1). We use scheduled sampling as mentioned in Section 4.1. We normalize the output embeddings and the ground truth embeddings to be unit vectors before computing the InfoNCE loss. The hyperparameters could be found in Appendix A.

**Comparison Systems** Off-the-shelf retrievers, trained on text inputs and outputs, will not produce meaningful performance on the synthetic dataset. Therefore, we train a single embedding model with comparable setting as our own model. This baseline outputs only a single query embedding and retrieves vectors from the corpus using that single output. We train this model using the standard InfoNCE loss, with the positive vector randomly sampled from the five targets. Each positive vector is paired with one negative vector. The other settings are the same as our system. We denote this baseline as **Single-Query**.

## 5.3 RESULTS

We show the results in Figure 3. While the single-query vector models never reach beyond MR @10 = 20%, the AMER model could perfectly (100%) retrieve all the target vectors in every setting. This validates our hypothesis: retrievers that output a single embedding cannot capture multimodal target distributions. By training the retriever on multiple target vectors, it learns to fit the predicted query embeddings to multiple distributions (in other words, learning multiple transformations).

The single-query model fails completely (0% across the board) on "Linear" data (orange bars). We believe this is because in the "Linear" data, two pairs of the transformations are negatives of each other, making it harder to learn. The single-query model is effectively learning to capture the average of the target vectors, and in the linear data the average of all target vectors is always zero. Along with the results in Section 3, we have shown the limitation of using a single query vector.

---

[4]The transformations stay the same; only the queries are sampled from out-of-distribution.

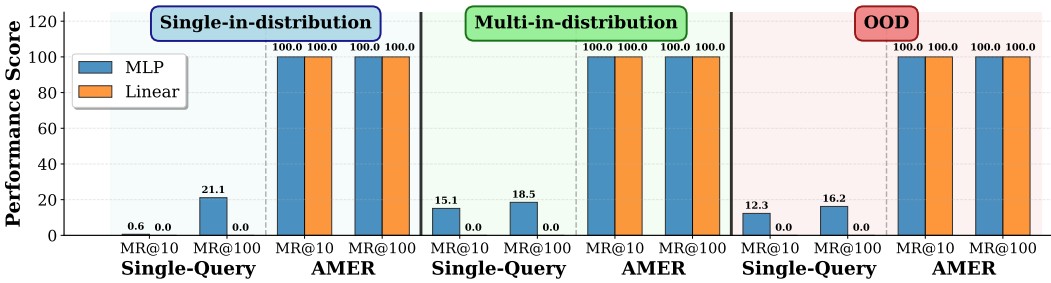

Figure 3: Results on synthetic data of Linear (Orange) and MLP (Blue) transformations. The y-axis represents performance scores, MRECALL @100 and 10. We evaluate systems on different input distributions, from a *Single* multivariate Gaussian, to *Multi*ple distributions as outlined and *OOD* distributions. Each section represents one input distribution. AMER (the right half of each section) can successfully model multiple target distributions, while the Single-Query (left) model struggles.

## 6 EXPERIMENTS: REAL-WORLD RETRIEVAL DATASETS

Finally, we evaluate the proposed method on real-world multi-answer retrieval datasets, AmbigQA (Min et al., 2020) and QAMPARI (Amouyal et al., 2023), as described in Section 2.

### 6.1 EXPERIMENTAL DETAILS

**Datasets** We report performances on the test split according to Table 1, and reserve 10% of the training data to be the validation set. To understand how models perform when the answers are more diverse (when the target document embeddings are less similar from one another), we additionally curate a subset (1/3) of the main test set where the cosine similarity between different targets of the same example is smaller (33 percentile). We denote this set as the "low similarity set". This corresponds to the right third of data region in Figure 1.

**Retrieval Desgin Choice** For efficiency in training and development, we fix the document encoder. Prior works have fixed the document encoder for simplicity of training (Vasilyev et al., 2025) or superior performances (Lin et al., 2023). While collecting hard negatives, which has been shown to impact model performances positively (Karpukhin et al., 2020; Xiong et al., 2021), we focus on whether the proposed AMER could mitigate the limitation of single-query embedding retrievers, not on achieving state-of-the-art performance on evaluation data, and test both the baselines and our model with random negatives. These design choices are orthogonal to our proposed architecture.

**Training Details** The retriever models have two components, query encoder and the document encoder. We use the Inf-Retriever (Yang et al., 2025b) as the document encoder.[5] We have the corpus embedding fixed throughout the training. For the query encoder model, we use various backbone LMs, including Llama-{1B,3B,8B} [6] and Qwen3-4B [7]. We train the model using LoRA fine-tuning (Hu et al., 2022) [8], with batch size of 128. Following Section 5, we use all the other positive documents in the same batch as in-batch negatives, do scheduled sampling and normalize the embeddings. Hyperparameter details are in Appendix A. During inference, we set a fixed number of query embeddings for each dataset ($N = 2$ for AmbigQA and $N = 5$ for QAMPARI).

### 6.2 BASELINES

We mainly compare to the **Single-Query** baseline as described in Section 5.2. We train this baseline using a randomly sampled target document as the positive in the contrastive objective, with

---

[5]We choose this model (`infly/inf-retriever-v1-1.5b` on HuggingFace) for its compact size (1.5B) and strong performance on MTEB (Muennighoff et al., 2023) leaderboard.

[6]`Llama-3.2-1B-Instruct`,`Llama-3.2-3B-Instruct`, and `Llama-3.1-8B-Instruct`

[7]`Qwen3-4B-Instruct-2507`

[8]We opt to do LoRA fine-tuning as the training set is relatively small, and full fine-tuning seems to drift the model too much from its base form.

Table 2: We report the relative performance gains ($\Delta$) on top of the Single-Query baseline, macro averaged across base LMs. We show that AMER outperforms the other baselines, and that the gain is much larger on subsets with lower pairwise similarity between target documents.

|  | AmbigQA (Whole Set) | AmbigQA (Low Similarity Set) | QAMPARI (Whole Set) | QAMPARI (Low Similarity Set) |
|---|---|---|---|---|
| Query Expansion | +2.89% | +1.27% | -13.50% | +2.39% |
| Re-ranking | +0.64% | +1.45% | -8.28% | +24.78% |
| AMER | +6.50% | +8.08% | +16.12% | +183.58% |

the exact same hyperparameters otherwise. To compare with traditional methods for improving diversity, we perform **Query Expansion** and **Re-ranking** on top of the single-query baseline. For query expansion, we ask GPT-4.1-mini to generate a few keywords, and append these keywords to the query. For re-ranking, we adopt the maximal marginal relevance objective (Carbonell & Goldstein, 1998) where the document scores are adjusted by a similarity penalty. We perform re-ranking on the top 500 retrieved document set. We detail these methods in Appendix A.7.

### 6.3 RESULTS

We present the results in Figure 4. AMER model shows consistent, modest gains over the Single-Query baseline. This indicates our multi-embedding objective is better at capturing multiple target distributions even in real-world settings, albeit by a smaller margin than observed in Section 5. We compare AMER with the Single-Query baseline using a paired bootstrap test, finding the performance gains are statistically significant in 6/8 settings on the entire test sets, and 7/8 settings on the low similarity sets. Two other baselines (query expansion and re-ranking) are ineffective in these datasets, showing marginal gains or even degradations. Yet, their performance, similar to ours, is stronger in low similarity set, showing diversity encouraging can hurt when target documents are similar to each other. We also

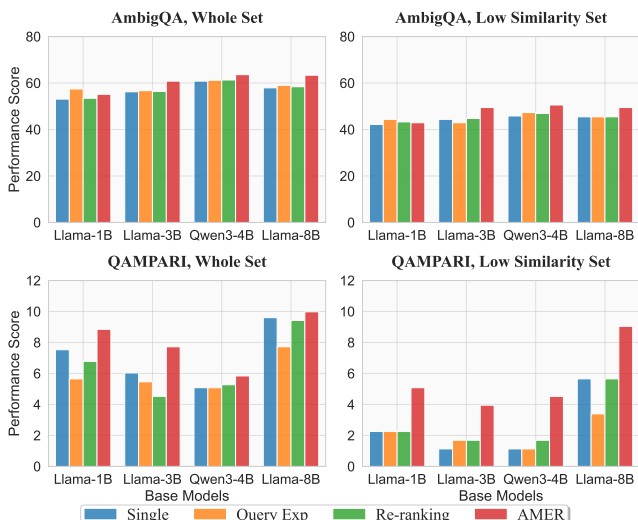

Figure 4: Performance for whole and low similarity test set for multiple base models. AMER outperforms baselines in most settings. In all models, we observe a stronger gain on the low similarity set. The gains are also larger on QAMPARI, which has a more diverse target distribution.

show the average relative performance gains ($\Delta$) across base LMs, computed as $\frac{B-A}{A}$, ($A$=Single-Query, $B$=Compared Systems) in Table 2. Query expansion and re-ranking baselines show limited gains on average compared to AMER. Detailed results are shown in Appendix A.8.

We observe that the performance gains compared to Single-Query baseline are larger on QAMPARI, possibly because the data is more diverse and the distance between target embeddings is larger. We hypothesize that our objective is more effective in scenarios where target distributions are more multimodal, or when the distance between target embeddings are larger, as we observed in the synthetic data. We find that in AmbigQA and QAMPARI, the average pairwise similarity among the gold documents of the same query (0.9 and 0.86, respectively) is substantially higher than their similarity to random documents from different queries (0.77 and 0.74). This indicates that the gold documents associated with each query are relatively homogeneous, which helps explain the modest performance gains we observe in practice. More detailed comparison between the diversity of real-world and synthetic datasets can be found in Appendix A.4.

**Output Embedding Diversity Analysis** We quantify output embedding diversity by measuring the average pairwise similarity between output embeddings for each query. We present the results in

Figure 5. Except for Llama-1B, retrievers trained using all the other base LMs show higher output diversity than the target distribution (Blue). This result demonstrates that AMER learns to output multiple distinct embeddings. We also find that larger models tend to produce more diverse embeddings.

## 6.4 ABLATION STUDY

Instead of doing scheduled sampling as mentioned in Section 4.1, we also experiment with always taking previously predicted embeddings as input during training ("Always Predicted"). We present the relative gains to the Single-Query baseline averaged across base LMs in Table 3. Doing scheduled sampling yields overall better results on AmbigQA.

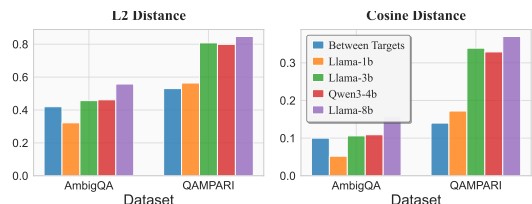

Figure 5: Vector similarity between multiple query embeddings from AMER, and the training data. "Between Targets" denotes the pairwise distance between target embeddings in the training dataset. Larger models exhibit overall higher diversity.

## 7 RELATED WORK

**Agentic Retrieval**  Another path to recover diverse information is to iteratively retrieve and refine the query to obtain new results. For example, Trivedi et al. (2023) proposes to interleave retrieval and Chain-of-thought reasoning to obtain new retrieval results. There are works in retrieval-augmented generation that iteratively uses newly generated queries to obtain new documents, either through prompting (Jiang et al.,

Table 3: Ablation Study on AmbigQA.

|  | Whole | Low Sim. |
| --- | --- | --- |
| Scheduled Sampling | +6.50% | +8.08% |
| Always Predicted | +6.01% | +6.46% |

2023; Li et al., 2025b;c) or a trained LM (Asai et al., 2024). More recent works train LMs with reinforcement learning (Jin et al., 2025; Song et al., 2025; Zheng et al., 2025; Chen et al., 2025). This line of research of query reformulation and building workflows with retrievers as a tool is orthogonal to our approach of improving the retriever architecture.

**Continuous Implicit Reasoning**  We make LMs generate the next embedding based on the previously predicted ones, without projecting the latent embeddings into discrete tokens. This is similar to a body of research where LMs "reason" in the latent space (Li et al., 2025a). Specifically, Cheng & Van Durme (2024) and Shen et al. (2025) propose to compress Chain-of Thought (CoT) in discrete tokens into a sequence of continuous embeddings. Hao et al. (2024) internalize the CoT to be continuous "thought" tokens in a multi-stage training process. Our method is different from these works since we actually use the generated embeddings as output, whereas they use them for deeper or more efficient reasoning but not shown as the final output.

**Multi-Query Vector Retriever**  A popular retriever architecture, ColBERT (Khattab & Zaharia, 2020), also generates multiple vectors per query. The motivation is to build a better token-level representation of the query, rather than modeling diverse outputs. A crucial difference is that ColBERT also represent each document with multiple vectors, and that it requires modeling interactions between these two sets of embeddings. This demands more expensive document indexing process and memory storage. A concurrent work (Weller et al., 2025) presents a theoretical result that for a fixed embedding dimension $d$, there exists a query relevance matrix that cannot be captured. We present a multi-query retriever that could address this issue.

## 8 FUTURE WORK AND CONCLUSION

We establish that single-query embedding retrievers cannot model diverse target distributions. Experiments on both synthetic and real-world data reveal that existing single-embedding retrievers exhibit worse performances on data with targets with larger distance in embedding space. We address this by proposing a multi-query vector approach. We establish its superior performance in retrieving diverse targets with empirical results on various datasets and model architecture. Future work could explore improvements to AMER, for instance learning to flexibly decide the number of query vectors to predict and more effectively aggregating across different outputs from different queries. Developing better training data, e.g. mining hard negatives and training on a much larger scale of unsupervised data, could scale this approach to more diverse tasks.

## 9 REPRODUCIBILITY STATEMENT

We describe the datasets we use in Section 2 and Section 5. We include the procedures to construct the data, and how we obtain the training and testing split. We also document more details in Appendix A.3 and A.2. We describe the model architecture and training objectives in Section 4. We describe the details for training, including hyperparameters in Section 5, 6 and Appendix A.5, A.6. We also document the models we use in Section 3, 5, and 6.

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

## A APPENDIX

### A.1 RETRIEVAL DIVERSITY DATASETS

Earlier works studied retrieval diversity in terms of disambiguating entities mentioned in the query (Clarke et al., 2008; Agrawal et al., 2009). We study queries that are associated with multiple target documents, specifically multi-answer datasets (Min et al., 2020; Amouyal et al., 2023). In this setting, evaluation focuses on the total recall of the retrieval model. Katz et al. (2023) have indexed named entity mentions in Wikipedia and showed that existing models are inadequate in capturing *all* relevant results. Chen & Choi (2025) studied retrieving diverse perspectives for subjective questions. Prior efforts for improving diversity focused on post-hoc methods like re-ranking (Carbonell & Goldstein, 1998) or preprocessing methods like query expansion. We propose to train an inherently diverse retriever that could capture multiple target distributions.

### A.2 DETAILS ON THE CONSTRUCTION OF SYNTHETIC DATA

**Input distributions.** The input queries are sampled from five distributions described below:

- Standard Gaussian $\mathcal{N}(0, I)$: The standard distribution mentioned in Section 5.1.

- High-variance Gaussian $\mathcal{N}(0, 4I)$: 4× larger variance for more spread-out queries.

- Correlated Gaussian: Sample a random positive definite covariance matrix for correlated dimensions.

- Uniform Distribution $[-2, 2]^d$: Uniform random vectors in a hypercube.

- Laplace + Gaussian: Sparse, spiky input vectors with noise. Laplace (double exponential) has heavier tails than Gaussian, meaning more extreme values. The distribution is then smoothed out by the small Gaussian noise.

We implement both the standard Gaussian $\mathcal{N}(0, I)$ and the high-variance Gaussian $\mathcal{N}(0, 4I)$ by calling the `numpy.random.multivariate_normal` function, using the `numpy` package. For the correlated Gaussian, we first create a symmetric and positive semi-definite matrix $AA^T$, where $A$ is a random matrix. We then let the covariance matrix of the multivariate Gaussian distribution to be $0.5 * AA^T + 0.1I$. For the Laplace + Gaussian, we implement the Laplace by calling `numpy.random.laplace(loc=0.0, scale=1.0, size=(n_input, d)` where n_input is number of input vectors, and $d$ is the vector dimension. We then add a Gaussian noise of mean=0 and variance =0.1.

**Generating the target vectors.** As explained, we apply either linear transformations or untrained MLPs to derive a diverse set of target vectors per query vector.

- **Linear transformations.** We let $T_1 = I$, which is the identity matrix, and $T_2 = -T_4 = M_a$, $T_3 = -T_5 = M_b$, where $M_a$ and $M_b$ are two random rotation matrices.

- **Multi-Layer Perceptron (MLP).** We design five two-layer Multi-Layer Perceptrons (MLPs) that share the same architecture, but initialized by different matrices. The weights of all the layers are $d$ by $d$. We use GeLU (Hendrycks & Gimpel, 2016) as the activation function. We create the initialization matrices using the following procedure:
    - Create a rotation matrix $M_a$.
    - Create another rotation matrix $M_b$ that is orthogonal to $M_a$ (in Frobenius inner product sense), and rotation matrix $M_c$ that is orthogonal to $M_b$ and $M_a$.
    - Let $M_d = -M_b$ and $M_e = -M_c$.

| Datasets | Question | Answers | Evidence Documents |
|---|---|---|---|
| AmbigQA | When did harry potter and the sorcerer's stone movie come out? | 4 November 2001, 16 November 2001 | **[1]** The film had its world premiere at the Odeon Leicester Square in London on **4 November 2001**, with the cinema arraged to resemble Hogwarts School. **[2]** The film was released to cinemas in the United Kingdom and the United States on **16 November 2001**. |
| QAMPARI | Who are the directors of movies produced by Eric Newman? | Zack Snyder, Ariel Schulman, and José Padilha | **[1]** Producers Eric Newman and MarcAbraham developed the film [...]. Dawn of the Dead is a 2004 American actionhorror film directed by **Zack Snyder** in hisdirectorial debut [...] **[2]** Project power is a 2020 American sciencefiction action film directed by Henry Jost and **Ariel Schulman**, produced by Eric Newman. **[3]** Newman conceived and produced [...]. Remakes of The Thing (2011) and Robocop(2014) followed [...]. Robocop is a 2014 American superhero filmdirected by **José Padilha**. |

Table 4: Example questions of both the AmbigQA and QAMPARI dataset. Both questions can be answered by multiple evidence document that suggests different valid answers.

And then we initialize the first MLP $T_1$ using $M_a$, $T_2$ using $M_b$, and so on. There are two weight matrices in each MLP, and we initialize both with the same matrix.

## A.3 DETAILS ON THE CONSTRUCTION OF REAL-WORLD DATA

We evaluate retrievers on a subset of the development set of both datasets. For AmbigQA, in order to obtain a training set, we spare a part of the dev set (59%) from AmbigQA and use the remaining as evaluation set. We further augment include multi-answer questions from the NQ-Open dataset (Lee et al., 2019) to form larger training set. For in-distribution evaluation, we evaluate on the subset of development set of QAMPARI where there are five to eight target documents.

AmbigQA and QAMPARI are datasets with multiple valid answers. We present one example of each dataset in Table 4. Both AmbigQA and QAMPARI do not provide ground truth evidence documents, and we need map the answers to ground truth documents in the corpus. We use the corpus from prior work (Amouyal et al., 2023), which is sourced from a Wikipedia dump from August 1st, 2021. Each document contains 100 words on average.

**AmbigQA** For AmbigQA, we first map evidence documents for the test data. We retrieve top 500 documents using BM25, Contriever, and `infly/inf-retriever-v1-1.5b`. We iterate over the union of the retrieval results and see if these documents contain any of the answers. If there existing a substring $s$ in document $d$ that exactly matches an answer $a$ ($s = a$), we refer to $d$ as a"gold document" of $a$. If we can find at least one gold document for all the answers, we keep that example. We reserve 50% of these data from the development set as the test data, a total of 827 data points. We repeat that for the training data of AmbigQA (N=10036). To increase the amount of training data, we also consider a subset of NQ-Open (Lee et al., 2019), where we combine the multi-answer examples in the development set and the test set (N=2398). We repeat the "gold document" mapping

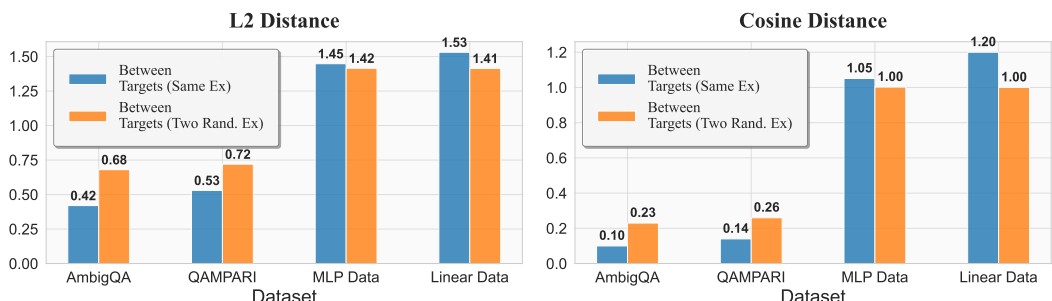

Figure 6: Comparing distance between (1) two target documents belonging to the same query and (2) two target documents from two randomly sampled queries. For an ideal dataset for evaluating diversity, the former (the blue bar) should be larger or equal to the latter (the orange bar). This is true for the synthetic datasets, but not for the real-world data.

process for this subset of NQ-Open, and we finally combine the filtered data from the training set of AmbigQA, the remaining 50% of the development set from AmbigQA and the multi-answer subset of NQ-Open. This results in a training set of total number of 5044 examples.

**QAMPARI**  For QAMPARI, the authors provided a short snippet in the original Wikipedia document for each answer. We retrieve top 500 documents using BM25, with the short snippet as query. We iterate the retrieved documents to find "gold documents" for the answers, following the procedure of AmbigQA. We keep the data point if we can find at least one "gold document" for all the answers. We only keep questions with five to eight answers. After these filtering steps, we retain 52% of the training set for training, and 53% of the development set for testing. The original sizes of the training and development set are 61911 and 1000, respectively.

### A.4 DETAILS ON THE TARGET DIVERSITY FOR SYNTHETIC VS. REAL DATA

We quantify diversity of the target document distributions by computing the pairwise distance between (1) two target documents of the same query and (2) two random target documents from two randomly sampled queries. We observe that distance between targets of the same query is significantly larger in synthetic datasets. This partly explains the smaller gain in real-world data compared to synthetic. Along with the results in Figure 4, we could infer that the performance gain from using our model is positively correlated with the distance between target embeddings from the same example.

### A.5 TRAINING DETAILS OF SYNTHETIC DATA EXPERIMENTS

**Model Architecture**  We use `Llama-3.2-1B-Instruct` as the base retriever model, and we do not need a document encoder as the data is already in vector form. The input and output linear layers are randomly initialized, and are of size $2048 \times 1024$. The vector data are of size 1024.

**Training Hyperparameters**  We train the model from base `Llama-3.2-1B-Instruct`. We split the training set into 90% training and 10% for validation. We perform hyperparameter tuning with learning rate of {1e-5, 2e-5, 5e-5, 1e-4}, per device batch size of {16, 32, 128}, temperature of {0.05, 0.04, 0.03, 0.02}, and number of epochs {500, 1000, 2000, 3000}. We gather batches from four GPUs, so the effective batch size would be 4X the per device batch size. We train the model with full fine-tuning, using AdamW optimizer (Loshchilov & Hutter, 2019) with $\beta_1$=0.9 and $\beta_2 = 0.98$. We use a linear learning rate scheduler with warmup rate = 0.05. We save checkpoints every 500 steps, and keep the one with lowest loss on the validation set.

### A.6 TRAINING DETAILS OF REAL-WORLD DATA EXPERIMENTS

**Base Models Used**  We use `Llama-3.2-1B-Instruct` as the base retriever model. We use `infly/inf-retriever-v1-1.5b` (Yang et al., 2025b) as the document encoder. We fix the document encoder during the whole training process.

**Training Hyperparameters** We largely follow the training recipe of the synthetic data experiments. The main difference is we are training models using LoRA fine-tuning (Hu et al., 2022), as we observe that the models deviate too much from its base form and yield worse performance when fully fine-tuned. We use LoRA rank of 64, LoRA alpha of 16, and LoRA dropout of 0.1. We apply LoRA to all the major linear layes ("q_proj,k_proj,v_proj,o_proj,down_proj,up_proj,gate_proj"). We perform hyperparameter tuning with learning rate of {1e-5, 2e-5, 5e-5, 1e-4}, per device batch size of {8, 16, 32}, temperature of {0.05, 0.04, 0.03, 0.02}, and number of epochs {30, 60, 120}. The other hyperparameters follow Appendix A.5.

## A.7 DETAIL OF BASELINES

**Query Expansion** For query expansion, the exact prompt we use is as follows:

> Write a list of keywords for the given question. The goal is to help retrieving relevant documents to the question, which contains multiple answers, so generate keywords as diverse as possible. Do not generate similar keywords; they should be distinct. Just answer with the list, and do not generate anything else. Keywords are separated by commas.
> Question: [Question]

where [Question] is replaced with the actual query. We use GPT-4.1-mini to generate the keywords. The rewritten query is the concatenation of the original query and the generated keywrords.

**Re-ranking** For re-ranking, we adopt the maximal marginal relevance (MMR) objective (Carbonell & Goldstein, 1998). We would like to maximize the below objective every time we add a new candidate document to the retrieval results:

$$\underset{D_i \in R \setminus S}{\arg\max}[\lambda Sim_q(D_i, Q) - (1 - \lambda) \underset{D_j \in S}{\max} Sim_d(D_i, D_j)] \tag{3}$$

where $R$ denotes the set of top retrieved documents considered for re-ranking candidates ($|R| = 500$), and $S$ denotes the document set that is already selected. We tune hyperparameter $\lambda$ on the development set (10% of training set) of each dataset with values [0.5, 0.75, 0.9]. $Sim_q$ is the cosine similarity score between each document and the query. $Sim_d$ is the cosine similarity between two document embeddings. For both similarity, we use the Inf-Retriever (Yang et al., 2025b).

## A.8 COMPREHENSIVE RESULTS ON REAL-WORLD RETRIEVAL DATASETS

We present comprehensive results on real-world retrieval datasets in Tables 5, 6, 7, and 8.

Table 5: Results on real-world multi answer datasets, using Llama-8B as backbone LM. We report MRECALL @ 100, and $\Delta$, which represents the relative performance gain from the Single-Query baseline. We compute the performance gain using the formula $\frac{B-A}{A}$, ($A$=Single-Query, $B$=Compared Systems). We report results on the whole test set and the subset where similarity between target document embeddings is lower as described in Section 2. **Bolded** are the best performances in each subset. We compared Single-Query with AMER results using a paired bootstrap test, and asterisk* indicates the difference is statistically significant ($p < 0.05$).

| Dataset | Systems | Whole Set | | Low Similarity Set | |
|---|---|---|---|---|---|
| | | MREC. @ 100 | $\Delta$ | MREC. @ 100 | $\Delta$ |
| AmbigNQ | Single-Query | 57.92 | 0.00% | 45.45 | 0.00% |
| | +Query Expansion | 59.01 | +1.88% | 45.45 | 0.00% |
| | +Re-ranking | 58.40 | +0.83% | 45.45 | 0.00% |
| | (Ours) AMER | **63.36*** | +9.39% | **49.45*** | +8.80% |
| QAMPARI | Single-Query | 9.60 | 0.00% | 5.65 | 0.00% |
| | +Query Expansion | 7.72 | -19.58% | 3.39 | -40.00% |
| | +Re-ranking | 9.42 | -1.88% | 5.65 | 0.00% |
| | (Ours) AMER | **9.98** | +3.96% | **9.04*** | +60.00% |

Table 6: Results on real-world multi answer datasets, using Qwen3-4B as backbone LM. Notations are the same as Table 5. **Bolded** are the best performances in each subset. We compared Single-Query with AMER results using a paired bootstrap test, and asterisk* indicates the difference is statistically significant ($p < 0.05$).

| Dataset | Systems | Whole Set | | Low Similarity Set | |
|---|---|---|---|---|---|
| | | MREC. @ 100 | $\Delta$ | MREC. @ 100 | $\Delta$ |
| AmbigNQ | Single-Query | 60.82 | 0.00% | 45.82 | 0.00% |
| | +Query Expansion | 61.19 | +0.61% | 47.27 | +3.16% |
| | +Re-ranking | 61.31 | +0.81% | 46.91 | +2.38% |
| | (Ours) AMER | **63.60*** | +4.57% | **50.55*** | +10.32% |
| QAMPARI | Single-Query | 5.08 | 0.00% | 1.13 | 0.00% |
| | +Query Expansion | 5.08 | 0.00% | 1.13 | 0.00% |
| | +Re-ranking | 5.27 | +3.74% | 1.69 | +49.56% |
| | (Ours) AMER | **5.84** | +14.96% | **4.52*** | +300.00% |

Table 7: Results on real-world multi answer datasets, using Llama-3B as backbone LM. Notations are the same as Table 5. **Bolded** are the best performances in each subset. We compared Single-Query with AMER results using a paired bootstrap test, and asterisk* indicates the difference is statistically significant ($p < 0.05$).

| Dataset | Systems | Whole Set | | Low Similarity Set | |
|---|---|---|---|---|---|
| | | MREC. @ 100 | $\Delta$ | MREC. @ 100 | $\Delta$ |
| AmbigNQ | Single-Query | 56.23 | 0.00% | 44.36 | 0.00% |
| | +Query Expansion | 56.71 | +0.85% | 42.91 | -3.27% |
| | +Re-ranking | 56.35 | +0.21% | 44.73 | +0.83% |
| | (Ours) AMER | **60.82*** | +8.16% | **49.45*** | +11.47% |
| QAMPARI | Single-Query | 6.03 | 0.00% | 1.13 | 0.00% |
| | +Query Expansion | 5.46 | -9.45% | 1.69 | +49.56% |
| | +Re-ranking | 4.52 | -25.04% | 1.69 | +49.56% |
| | (Ours) AMER | **7.72*** | +28.03% | **3.95*** | +249.56% |

Table 8: Results on real-world multi answer datasets, using Llama-1B as backbone LM. Notations are the same as Table 5. **Bolded** are the best performances in each subset. We compared Single-Query with AMER results using a paired bootstrap test, and asterisk* indicates the difference is statistically significant ($p < 0.05$).

| Dataset | Systems | Whole Set | | Low Similarity Set | |
|---|---|---|---|---|---|
| | | MREC.
@ 100 | Δ | MREC.
@ 100 | Δ |
| AmbigNQ | Single-Query | 53.08 | 0.00% | 42.18 | 0.00% |
| | +Query Expansion | **57.44** | +8.21% | **44.36** | +5.17% |
| | +Re-ranking | 53.45 | +0.70% | 43.27 | +2.58% |
| | (Ours) AMER | 55.14* | +3.88% | 42.91 | +1.73% |
| QAMPARI | Single-Query | 7.53 | 0.00% | 2.26 | 0.00% |
| | +Query Expansion | 5.65 | -24.97% | 2.26 | 0.00% |
| | +Re-ranking | 6.78 | -9.96% | 2.26 | 0.00% |
| | (Ours) AMER | **8.85*** | +17.53% | **5.08*** | +124.78% |

