# OpenReview forum: "Beyond Single Embeddings: Capturing Diverse Targets with Multi-Query Retrieval"
_ICLR.cc/2026/Conference — Submitted to ICLR 2026_

### Official Review · Reviewer_Q6bx · 2025-10-29

**Soundness:** 2
**Presentation:** 2
**Contribution:** 2
**Rating:** 2
**Confidence:** 4

**Summary:**

This paper proposes the Autoregressive Multi-Embedding Retriever (AMER) to address the limitation of standard dense retrievers, which typically generate a single query embedding and struggle to retrieve diverse sets of relevant documents for queries with multiple valid answers. AMER generates multiple query embeddings per query autoregressively, enabling better retrieval of diverse targets. Experiments on both synthetic and real-world datasets show that AMER significantly outperforms single-embedding baselines, especially when target documents are less similar to each other, demonstrating the importance and effectiveness of modeling diversity in document retrieval.

**Strengths:**

+ This paper proposes a new retrieval method where the original text, processed through the proposed AMER framework, can generate multiple retrieval embeddings. These embeddings enable comprehensive retrieval of the original text from multiple dimensions, identifying multiple documents containing the correct answers and thereby improving the performance of multi-question retrieval.
+ The effectiveness of the method is further validated through experiments.

**Weaknesses:**

+ Indeed, the idea of representing queries and documents using multiple vectors for retrieval, namely multi-vector retrieval, is not new, and the authors seem to ignore such relevant literature. One notable method that the authors need to mention and compare against is "Colbert: Efficient and effective passage search via contextualized late interaction over bert, SIGIR 20", which is the first paper on multi-vector retrieval. The authors need to cite and compare against this paper.
+ I understand that the authors may want to learn multiple query embeddings for retrieval, which is not explored by papers like ColBert. However, one recent paper, "POQD: Performance-Oriented Query Decomposer for Multi-vector retrieval, ICML 25" has proposed a relevant idea, i.e., dynamically obtaining these query embeddings through learning to decompose queries. The authors also need to analyze the difference from this work and ideally compare the proposed method against it in experiments.
+ Some technical details are also not clear to me, particularly, how the document embeddings are obtained. Based on my understanding, each document is also represented by multiple embeddings, which play a critical role in learning query embeddings. However, no description is provided on how these document embeddings are generated for each document.

**Questions:**

See above

---

> ### Author Response · Authors · 2025-11-26
>
> We thank the reviewer for their time and effort. We address the questions below:
>
> **Re: Comparison to ColBERT**
> See **Re:ColBERT**.
>
> **Re: Related Work - Performance-Oriented Query Decomposer**
> Thank you for suggesting relevant work! We missed it as it was focused on image retrieval rather than text. This work focuses on refining query decomposition based on ColBERT architecture, rather than ColBERT’s original decomposition based on the token. This work is not directly comparable, as it also introduces multiple document embeddings. We will include related work in the revised manuscript.
>
>
> **Re: How to obtain document embeddings**
> We report in L191 that document embeddings are generated through the fixed document encoder (pre-trained off-the-shelf retriever). The document embedding is a single vector, which is of the same dimension of the query embedding. The document encoder takes a sequence of text as input and outputs a single vector. It does not produce multiple document embeddings, which is a major difference from ColBERT and POQD.
>
> Reference:
> 1. Liu, Yaoyang, Junlin Li, and Yinjun Wu. "POQD: Performance-Oriented Query Decomposer for Multi-vector retrieval." In Forty-second International Conference on Machine Learning.

---

### Official Review · Reviewer_XqTN · 2025-10-31

**Soundness:** 3
**Presentation:** 3
**Contribution:** 2
**Rating:** 4
**Confidence:** 3

**Summary:**

The paper tackles the task of multi-query retrieval. It introduces AMER, a retrieval model that generates multiple query embeddings autoregressively. All the predicted query vectors are then used to retrieve documents. This approach directly addresses the limitations of
single-query vector retrievers and enables retrieving diverse outputs. Finally, the authors test the proposed approach on several benchmarks (one synthetic introduced in the paper and two other existing benchmarks).

**Strengths:**

The paper tackles an important task. It shows some limitations in current methods and proposes a method that addresses them. Overall, the idea seems interesting to me, but I feel like the improvement on real world tasks is a lot lower than I expected.

**Weaknesses:**

My main concern is related to the experimental part: while I agree that in principle one query can be mapped to diverse documents and the experiments show that there's a degradation in performance by choosing the most diverse set of documents, the synthetic data experiments feel tailored explicitly for this case. If the proposed method achieves 100% performance on the synthetic data, it feels like the experiment is either to simple or specifically tailored for this case. I think that overall it is true that you can in principle have a query matched to diverse documents, but there's a limit to that - if the set of documents is too diverse, then the query shouldn't retrieve all of them. So, I feel like there's no real insight gained from the synthetic data experiment. My next concern is related to the novelty which seems relatively limited since the idea of learning multiple embeddings for the same query has been explored for example in cross-modal retrieval

Yale Song, Mohammad Soleymani; Proceedings of the IEEE/CVF Conference on Computer Vision and Pattern Recognition (CVPR), 2019, pp. 1979-1988

weak comparison with other methods (see below)

**Questions:**

Maybe I am missing something, but why isn't the sota comparison made against methods from Fig 1?

---

> ### Author Response · Authors · 2025-11-26
>
> Thank you for the careful review. We address the reviewer’s concerns below:
>
> **Re: Contribution of the synthetic experiments**
> We intentionally designed a simplified synthetic experiment setting to show the following: (1) empirically show the inherent limitation of a single-query retriever; it cannot retrieve targets that are dissimilar. (2) given a diverse target distribution, the proposed AMER architecture can capture all the targets.
>
> After seeing initial positive results with synthetic dataset, we moved on to real-world dataset, as they were our main goal to begin with. Having said that, we could have designed harder synthetic datasets to understand the model’s ability further. For example, we could introduce more complex data transformations than linear / simple MLP layers, or train adversarial data generators (i.e., find K linear transformations that’d be challenging for AMER). We could also have introduced a varying number of target distributions per query. Given the performances on real world datasets, where we already see challenges, we did not think it’s necessary to design more complex synthetic datasets.
>
>
> **Re: Related work in cross-modal retrieval**
> Thank you for the pointer! The mentioned related work [1, Song et al., 2019] in cross-modal retrieval investigates the multiple instance learning (MIL) problem, which tackles tasks where the training examples are ambiguous, thus the goal of the paper is different from ours. In their case, only one of those feature vectors is responsible for the observed classification. In our case, all of the feature vectors are used and mapped to diverse target distributions.
> The other difference is that for them, multiple vectors are learned with different local guided features, in parallel. However, for us, multiple query vectors are generated autoregressively (sequentially). Newly generated vectors depend on previously generated ones, avoiding duplicates. We will add discussions in the related work.
>
>
> **Re: Small performance improvements**
> It is not easy to obtain performance gain on information retrieval benchmarks. For example, on the well-known MTEB benchmark [2, Muennighoff et al., 2023], the absolute performance difference of mean score on the benchmark is often less than 1 point between top models.
>
> For example, in the paper suggested by reviewer AiVt [3, Kuo et al., 2025], the average relative performance gain on top of the best baseline (other than the proposed model) in Table 1 is 4.5% in terms of Recall @ 1k, and 5.5% in terms of nDCG @ 10.
>
> In related work in cross-modal retrieval[1, Song et al., 2019], the relative performance gain of the proposed method on top of the best baseline in Table 2 is -0.04% on Recall @ 1, 0.1% on Recall @ 5, and -0.23% on Recall @ 10.
>
>
> Although our method (AMER) only shows modest relative performance gain (shown in Table 2 in the paper) on AmbigQA over the single-query baseline (6.5%), this is comparable to listed previous works. Our method also shows larger relative performance gains compared to the single-query baseline on QAMPARI (>15% gain), and an even larger gain on the low similarity subset of QAMPARI (>100% gain).
>
>
> **Re: SOTA model comparison**
> We start the training (of the query encoder) of our model (AMER) and the single-query baseline from a pretrained base LM. The models shown in Figure 1. are trained on much more data, including text embedding tasks and retrieval tasks. The comparison would not be very meaningful, as models are starting from drastically different starting points. We aim to show that the AMER model could obtain improvements from the single-query baseline, but not to claim that we have achieved SOTA performance.
>
>
> Reference:
> 1. Song, Yale, and Mohammad Soleymani. "Polysemous visual-semantic embedding for cross-modal retrieval." Proceedings of the IEEE/CVF conference on computer vision and pattern recognition. 2019.
> 2. Muennighoff, Niklas, et al. "Mteb: Massive text embedding benchmark." Proceedings of the 17th Conference of the European Chapter of the Association for Computational Linguistics. 2023.
> 3. Yuan-Ching Kuo, Yi Yu, Chih-Ming Chen, and Chuan-Ju Wang. 2025. MMLF: Multi-query Multi-passage Late Fusion Retrieval. In Findings of the Association for Computational Linguistics: NAACL 2025, pages 6587–6598, Albuquerque, New Mexico. Association for Computational Linguistics.

---

### Official Review · Reviewer_AiVt · 2025-11-03

**Soundness:** 2
**Presentation:** 3
**Contribution:** 2
**Rating:** 4
**Confidence:** 2

**Summary:**

This paper targets retrieval settings where a single query has multiple, diverse target documents and single-embedding retrievers fail to cover distant clusters. It proposes AMER, an autoregressive multi-embedding retriever that generates a sequence of query vectors from the query encoder while keeping the document encoder frozen. Training uses InfoNCE with Hungarian matching to align unordered positives to generated vectors and scheduled sampling to reduce exposure bias. At inference, each vector retrieves candidates that are merged via round-robin. Results show perfect coverage on synthetic data and consistent, sometimes large, gains on AmbigQA/QAMPARI, especially when targets are dissimilar.

**Strengths:**

This paper reframes retrieval for multi-target queries as autoregressive generation of multiple query embeddings, creatively combining sequence modeling with contrastive retrieval. It uses Hungarian matching to align unordered positives and scheduled sampling to reduce exposure bias, removing a core limitation of single-vector retrievers.

**Weaknesses:**

(1) Baselines are incomplete. The paper does not compare against strong multi-vector retrievers (e.g., the ColBERT[1] family) or competitive query-rewriting/expansion approaches (such as MMLF[2]). Although architectural limitations of late interaction are mentioned, the lack of head-to-head metrics under matched budgets makes it hard to gauge relative advantage.

(2) Single-answer regimes are underexplored. It remains unclear how the method behaves when a query has a single narrow intent.

(3) The work does not explore instruction-tuning that ask the model to produce multiple distinct embeddings , leaving open whether instruction control could better leverage LLM generative priors for diversity.

[1]. Omar Khattab and Matei Zaharia. Colbert: Efficient and effective passage search via contextualized
late interaction over bert. In Proceedings of the 43rd International ACM SIGIR conference on
research and development in Information Retrieval, 2020.

[2]. Yuan-Ching Kuo, Yi Yu, Chih-Ming Chen, and Chuan-Ju Wang. 2025. MMLF: Multi-query Multi-passage Late Fusion Retrieval. In Findings of the Association for Computational Linguistics: NAACL 2025, pages 6587–6598, Albuquerque, New Mexico. Association for Computational Linguistics.

**Questions:**

As above.

---

> ### Author Response · Authors · 2025-11-26
>
> Thank you for the thoughtful feedback. We address the reviewer’s questions below:
>
> **Re: Incomplete Baselines**
> See **Re: ColBERT** results for comparing against ColBERT.
>
> **MMLF [1, Kuo et al., 2025] results:**
> Thanks for suggesting another baseline. We have implemented the baseline as below and report the comparison with our method. We reproduce their setup by using “meta-llama/Meta-Llama-3-70B” as the query rewriting and document generation model. We also follow the original implementation and use “Reciprocal rank fusion” for aggregating the retrieved results from the original query and $n$ rewritten queries. We adopt $n=2$ for AmbigQA and $n=5$ for QAMPARI, which follows the number of queries generated by our model (AMER). We use the single-query baseline as the retriever for fair comparison. We show the relative performance gain (with MRecall @ 100 as metric) on top of the single-query baseline macro averaged across base LMs, following Table 2 in our paper. The other rows are copied from the paper.
>
> |                      | **AmbigQA (Whole Set)** | **AmbigQA (Low Similarity Set)** | **QAMPARI (Whole Set)** | **QAMPARI (Low Similarity Set)** |
> |--------------------|-------------------|------------------------|-------------------|------------------------|
> | **Query Expansion**  | +2.89%         | +1.27%              | -13.50%         | +2.39%               |
> | **Re-ranking**       | +0.64%              | +1.45%              | -8.28%           | +24.78%              |
> | **MMLF**             | **+0.52%**         | **+0.96%**          | **-15.19%**    | **-23.71%**          |
> | **AMER**             | +6.50%              | +8.08%                | +16.12%        | +183.58%            |
>
>
> One hypothesis why MMLF did not improve performance is that the performance largely depends on whether the generated documents are relevant to the question. It is likely that the questions in QAMPARI are less answerable with LMs’ parametric knowledge than the question in the original MMLF paper (which is the BEIR dataset, [2, Thakur et al., 2021]). This might also explain why MMLF slightly helps AmbigQA.
>
>
>
> **Re: Single-answer regimes**
> This is a valid concern. Below, we present additional experimental results on a subset of the development set of the AmbigQA where there is only one answer. We compare our proposed approach (AMER) against the single-query baseline, and **AMER still outperforms the single-query baseline**. As expected, however, the performance gain is also smaller compared to the multi-answer portion we evaluate on in the paper.
>
> Below is the performance on the single-answer portion of AmbigQA using different base models.
> | Model & Method        | **Recall @ 100** | **Precision @ 100** |
> |------------------|------------------|-----------------|
> | Llama-1b, Single-Query | 85.93            | 10.49                |
> | **Llama-1b, AMER**     | **88.85**        | **11.14**            |
> | Llama-3b, Single-Query | 89.88            | 11.43                |
> | **Llama-3b, AMER**     | **91.67**        | **12.14**            |
> | Qwen3-4b, Single-Query | 88.66            | 11.69                |
> | **Qwen3-4b, AMER**     | **89.56**        | **11.97**            |
> | Llama-8b, Single-Query | 89.69            | 12.28                |
> | **Llama-8b, AMER**     | **92.88**        | **13.71**            |
>
>
> Below is relative performance delta (comparing AMER vs. single-query) on different datasets. We show the relative performance gain of AMER (with MRecall @ 100 as metric) on top of the single-query baseline macro averaged across base LMs, following Table 2 in our paper. The numbers in row 2 and 3 in the following table are copied from Table 2 in the paper.
> | Dataset                      | **Whole Set** | **Low Similarity Set** |
> |---------------|---------------|------------------|
> | **AmbigQA (Single-Answer)**  | **+2.48%**    | -                      |
> | AmbigQA (Multi-Answer)       | +6.50%        | +8.08%                  |
> | QAMPARI                      | +16.12%       | +183.58%                |
>
>
>
> **Re: instruction-tuning that ask the model to produce multiple distinct embeddings**
> Existing dense retriever models, even when instruction-tuned, are not capable of producing multiple embeddings, as they map an input text into an embedding. This is why our proposed architecture is novel.
>
> Reference:
> 1. Yuan-Ching Kuo, Yi Yu, Chih-Ming Chen, and Chuan-Ju Wang. 2025. MMLF: Multi-query Multi-passage Late Fusion Retrieval. In Findings of the Association for Computational Linguistics: NAACL 2025, pages 6587–6598, Albuquerque, New Mexico. Association for Computational Linguistics.
> 2. Thakur, Nandan, et al. "BEIR: A Heterogeneous Benchmark for Zero-shot Evaluation of Information Retrieval Models." Thirty-fifth Conference on Neural Information Processing Systems Datasets and Benchmarks Track (Round 2).

---

### Official Review · Reviewer_YjuD · 2025-11-04

**Soundness:** 3
**Presentation:** 3
**Contribution:** 2
**Rating:** 6
**Confidence:** 4

**Summary:**

This paper tackles a common and practical problem: standard search retrievers are bad at handling ambiguous queries. The authors first demonstrate that existing retrievers all struggle as the distance between target document embeddings grows and then go ahead to present AMER as a solution. Instead of generating just one query vector, it auto-regressively generates multiple query vectors. To handle the fact that the multiple target documents are an unordered set, the authors also employ an elegant Hungarian matching algorithm to find the optimal pairing between generated query embeddings and target document embeddings.

**Strengths:**

- The paper's core idea is simple, intuitive, and rigorously tested. It is also well written.
- The authors tackle a core, fundamental limitation of the dominant bi-encoder retrieval paradigm. By providing a practical architecture to move "beyond single embeddings", this work opens a new and important direction for retrieval model design.
- The authors look at performance gain on synthetic benchmark that showcases it's superior performance but also show the moderate gain on real world data.

**Weaknesses:**

- The authors state they "assume a setting" with a frozen document encoder for "faster development". This is a major experimental concession. While this makes testing easier, it creates a potential disconnect between the latest documents.
- There might be a potential practical downside of higher inference cost due to *m* separate matches.
- The true number of distinct answers (or answer clusters) varies per query, from one to many. This fixed parameter might not perform well in certain cases or might be inefficient.
- To train this system, you need to find all the gold documents for all the different answers for each query. This can be potentially noisy and unstable.

**Questions:**

- Have the authors though about any practical limitations like increased cost/latency or fixed document encoder?

---

> ### Author Response · Authors · 2025-11-26
>
> We thank the reviewer for their constructive comments. We appreciate that the reviewer finds our experiment rigorous and our paper well written. We address the reviewer’s questions below:
>
> **Re: Frozen document encoder**
> We chose to freeze the document encoder for implementation simplicity, as this requires re-indexing the whole corpus (which takes about 200 GPU hours) before inference. Prior works have also fixed the document encoder either for simplicity of training [1, Vasilyev et al., 2025] or superior performances [2, Lin et al, 2023].
>
> We are not sure what “potential disconnect between the latest documents” means, but one potential concern might be that the document indices are not updated when the training progresses, as some previous works update the document indices asynchronously. We would like to clarify that we do not update the document encoder at all, so the fixed document indices would not be a problem. Please let us know if this addresses your concern.
>
>
> **Re: Higher inference cost for having multiple queries**
> Thank you for raising the concern! We agree inference speed is crucial. We present the actual inference time for each setting (in seconds). We also include a more competitive query rewriting baseline (MMLF [5, Kuo et al., 2025]) per suggestion from Reviewer AiVt.
> We measured latency on H200s with 140GB memory, and we reported the average time taken for inference using Llama-1B and Llama-8B as base models. We divide the inference time into “query encoding” which is when the query embeddings are generated, and “lookup” which is when the nearest neighbor search happens.
> We use the “faiss-gpu” library for nearest neighbor search and we move the document indices to GPUs, so it does not take much time for an additional search query. Loading the index to GPUs takes more time than searching.
>
> Our method is more efficient than the query expansion / reranking approach, and only 36% more expensive than the single-query baseline. We will add these results and discussion to the revised draft.
>
> | Model      | Dataset          | **Single-Query** |      |      | **AMER** |      |      | **Query Expansion** |       |      |      | **Re-ranking** |      |        |        | **MMLF** |      |      |      |
> |------------|--------------|----------------|--------|--------|----------|--------|--------|-----------------|-------|------|------|----------------|--------|--------|--------|--------|--------|--------|--------|
> |            |            | Query Encode     | Lookup | **Total**  | Query Encode | Lookup | **Total**| Gen Keywords | Query Encode | Lookup | **Total**| Query Encode | Lookup | Re-ranking | **Total** | Gen N docs | Query Encode | Lookup | **Total** |
> | **Llama-1B** | QAMPARI (n=5)   | 16.25            | 541.72 | **557.97** | 46.65 | 666.93 | **713.58** | 1463.13   | 16.45     | 511.85 | **1991.43** | 16.25 | 541.72 | 10975  | **11532.97** | 608.36    | 16.31 | 544.12 | **1168.79** |
> |        | AmbigQA (n=2)  | 21.85   | 534.47 | **556.32** | 32.94   | 790.94 | **823.88** | 2329.25  | 19.07    | 531.13 | **2879.45** | 21.85        | 534.47 | 9260   | **9816.32** | 165.77    | 21.79        | 529.35 | **716.91** |
> | **Llama-8B** | QAMPARI (n=5)   | 36.80   | 420.88 | **457.68** | 119.98  | 548.40 | **668.38** | 1463.13    | 45.56    | 421.67 | **1930.36** | 36.80  | 420.88 | 11138  | **11595.68** | 608.36    | 32.17   | 434.25 | **1074.78** |
> |        | AmbigQA (n=2)    | 58.26  | 590.25 | **648.51** | 77.74  | 705.79 | **783.53** | 2329.25   | 72.80  | 604.08 | **3006.13** | 58.26        | 590.25 | 9362   | **10010.51** | 165.77    | 58.3   | 592.17 | **816.24** |
>
>
> **Re: Predict fixed number of query embeddings**
> We agree that generating a flexible number of query embeddings based on the query semantics would be great! We have explored this earlier in our study, by (1) training the model to generate [eos] embedding at the end of generation, and (2) first generating the number of embeddings and then generating that number of embeddings. However, it was challenging to stably train a model under this setting. This would be a great direction for future work, and we will discuss this in the revised version.
>
> **Re: The need to find all gold documents**
> Finding the gold documents is necessary for training other dense retriever models, so this is not a unique weakness specific to our proposed retriever architecture. Having a comprehensive set of answers during training is not a strict requirement, and even a subset of answers should be sufficient to train. Future work could explore distance supervision methods to mine “silver” documents to form the positives. Another potential direction is to generate large-scale synthetic data that naturally has multiple diverse targets, just like how dense embedding models were initially trained with supervised data (DPR: [3, Karpukhin et al, 2020]) to unsupervised data (Contriever: [4, Izacard et al, 2022]).
>
>
> (Continued for reference)

---

> > ### Author Response · Authors · 2025-11-26
> > **Reference:**
> >
> > Reference:
> > 1. Vasilyev, Oleg, Randy Sawaya, and John Bohannon. "Preserving multilingual quality while tuning the query encoder on English only." NAAACL. 2025. (https://aclanthology.org/2025.naacl-short.28/)
> > 2. Lin, Xi Victoria, et al. "Ra-dit: Retrieval-augmented dual instruction tuning." ICLR 2023. (https://arxiv.org/abs/2310.01352)
> > 3. Vladimir Karpukhin, Barlas Oguz, Sewon Min, Patrick Lewis, Ledell Wu, Sergey Edunov, Danqi Chen, and Wen-tau Yih. 2020. Dense Passage Retrieval for Open-Domain Question Answering. In Proceedings of the 2020 Conference on Empirical Methods in Natural Language Processing (EMNLP), pages 6769–6781, Online. Association for Computational Linguistics.(https://aclanthology.org/2020.emnlp-main.550/)
> > 4. Izacard, Gautier, Mathilde Caron, Lucas Hosseini, Sebastian Riedel, Piotr Bojanowski, Armand Joulin, and Edouard Grave. "Unsupervised Dense Information Retrieval with Contrastive Learning." Transactions on Machine Learning Research. (https://openreview.net/pdf?id=jKN1pXi7b0)
> > 5. Yuan-Ching Kuo, Yi Yu, Chih-Ming Chen, and Chuan-Ju Wang. 2025. MMLF: Multi-query Multi-passage Late Fusion Retrieval. In Findings of the Association for Computational Linguistics: NAACL 2025, pages 6587–6598, Albuquerque, New Mexico. Association for Computational Linguistics.

---

### Author Response · Authors · 2025-11-26
**Re: Comparison with ColBERT results**

We discuss the ColBERT in the related work section (L467-474). ColBERT is a compact dense retriever model that decomposes both the query and the document into tokens and obtains their token-level representations. Similarity between query and documents is then computed via a “late interaction” operation, which is a sum over maximum similarity, $$Sq,d :=\sum_{i ∈[ |Eq |]}\max_{j ∈[ |Ed |]} E_{q_i} · E^T_{d_j}$$, where $E_{q_i}$ and $E_{d_j}$ are token-level query and document embedding, respectively.

**Except that both our work and ColBERT generate multiple query embeddings per query, there’s a little similarity between ours and ColBERT.** Our goal is to capture diverse high-level semantics and answer spaces of the query, while in ColBERT, is to represent each token in the query with an embedding. Therefore, in our work, the number of tokens per query in our approach is a hyperparameter (would be great to be able to learn it as a future work!), while in ColBERT, is not changeable, fixed as the number of tokens in the query.

Reviewer AiVt asked for a head-to-head comparison with ColBERT, but we found it pretty challenging for the following reasons:

1. Efficiency:
We adopt a single-document embedding scenario. This is the standard setting where most retrieval models are compared at MTEB benchmark [3, Muennighoff et al., 2023]. In all these, the number of embedding per document is 1. In ColBERT, the number of embeddings per document is the number of tokens in the documents.
This impacts efficiency in two dimensions (A) the size of document corpus index and (B) the inference time latency.

    (A) The size of document corpus index: naively implemented, ColBERT index would be as large as $m$ times the single embedding approaches, where $m$ is the average length of the document in the corpus. Yet, ColBERTv2 [1, Santhanam et al., 2022]  has introduced many optimization techniques (e.g., compression) to bring down this cost. However, such optimization techniques are not specific to multi-document settings and can be theoretically applied for single embedding document cases (like ours) as well. Even after such optimizations, ColBERT incurs larger memory footprint (1.66X larger than base InfRetriever (document index we are using)) while using a smaller embedding dimension ($d$=768) instead of ($d$=1536).

    (B) The inference time latency: If the queries contain on average $n$ tokens and the documents contain on average $m$ tokens, there is $m*n$ times the computation, compared to $K$ (the number of query embeddings we generate per query) computation needed for our approach.

2. Difference in training data:
ColBERT was initialized with a base LM and then trained on large-scale data (MS MARCO [2, Bajaj et al., 2016], ~1M queries), whereas our model is initialized from a base LM and only fine-tuned with a small scale training data (AmbigQA and QAMPARI, 10K and 32K respectively). It is a bit hard to do head-to-head comparison with different training dataset conditions. Our setting is easier in that it is in-domain data, but they benefit from training on a substantially larger scale, still fairly similar domain data.


Because of these differences, we do not think ColBERT should be considered as a baseline.
Having said that, we provide the comparison with ColBERT below, as a datapoint, using the pretrained ColBERT-v2 checkpoint provided by the authors. We report MRecall @ 100 following our paper.

1. AmbigQA.
| Model       | **Whole Set**  | **Low Similarity Set**   | **Index Size** |
|------------------|----------------|----------------|----------------|
| **ColBERT-v2**   | 75.94      | 66.55   | 101 GB         |
| **AMER (Llama-8B)** | 63.36        | 49.45       | 61 GB          |


2. QAMPARI

| Model            | **Whole Set**   | **Low Similarity Set**          | **Index Size** |
|------------------|----------------|-------------------------|----------------|
| **ColBERT-v2**   | 34.84         | 21.47      | 101 GB      |
| **AMER (Llama-8B)** | 9.98        | 9.04   | 61 GB          |



ColBERT-v2 does outperform our model. The superior performance could come from larger expressivity of query and documents, pretraining objectives, or pretraining data.

We also would like to emphasize that **our goal was not to create a SOTA model on AmbigQA and QAMPARI, but to show that retrievers that output multiple query vectors could outperform their single-query counterpart with careful optimization.** We also show the limitation in existing single-query retrievers and the potential of multi-query embedding retrievers. We’d add these results and discussion into the revised draft.

---

### Meta-Review · Area_Chair_HvAr · 2026-01-04

**Summary:**

The reviewers reached a consensus that while the paper explores an interesting direction in multi-vector retrieval, it is currently premature for publication. The main critical concerns involve 1) limited novelty and insufficient literature coverage, 2) an incomplete experimental evaluation against established state-of-the-art baselines, and 3) significant methodological concessions. In the rebuttal, the authors provided new results for ColBERT and MMLF; however, these results showed that the proposed method (AMER) is outperformed by existing frameworks like ColBERT-v2. Furthermore, the authors' justification for certain design choices—such as a frozen document encoder and a fixed number of query embeddings—highlights that the work remains a proof-of-concept rather than a robust, competitive retrieval system. Because the rebuttal confirmed that the method does not yet achieve competitive performance and leaves major technical questions (e.g., dynamic query counts) to future work, the AC recommends rejection of the paper.

**Reviewer Concerns:**

- Literature Coverage and Novelty: The authors acknowledge missing key literature, including ColBERT and POQD. While the rebuttal attempts to distinguish AMER by its use of autoregressive high-level semantic embeddings versus token-level embeddings, the core idea of multi-vector representation is already explored. The failure to engage with these established frameworks in the original submission limited the paper's ability to clearly define its unique contribution. Furthermore, another missing line of relevant works is the multi-hop retrievers such as ReSCORE (Lee et. al., 2025), which significantly resemble the proposed method.
- Comparison to Baselines: The rebuttal included a "head-to-head" comparison with ColBERT-v2 and MMLF, but the results were not favorable. The authors admit that ColBERT-v2 outperforms their model, attributing this to "larger expressivity" and "pretraining data." While the AC appreciates the new data, it confirms the reviewers' suspicion that the proposed method is not yet competitive with existing multi-vector or query-rewriting solutions.
- Experimental and Technical Concessions: Several design choices were criticized as overly restrictive. The authors confirm they used a frozen document encoder for "implementation simplicity," which reviewers noted creates a potential disconnect between query and document representations. Additionally, the authors acknowledge that predicting a fixed number of embeddings regardless of query intent is a limitation they could not resolve stably, leaving a key piece of the proposed motivation unaddressed.
- Synthetic Data and Insights: Reviewers found the synthetic experiments to be overly simplified (achieving 100% performance). The authors' response—that they did not think complex synthetic tests were necessary—does not address the reviewers' concern that these experiments provide little insight.

**Reviewer Scores:**

As the key concerns remain unaddressed, the AC anticipates that the lone 'Borderline Accept' would likely shift toward a negative evaluation, solidifying a clear negative consensus among the reviewers.

---

### Decision · Program_Chairs · 2026-01-26

Reject